# Leveraging heterogeneous spillover in maximizing contextual bandit rewards

## Abstract

Recommender systems relying on contextual multi-armed bandits continuously improve relevant item recommendations by taking into account the contextual information. The objective of bandit algorithms is to learn the best arm (e.g., best item to recommend) for each user and thus maximize the cumulative rewards from user engagement with the recommendations. The context that these algorithms typically consider are the user and item attributes. However, in the context of social networks where *the action of one user can influence the actions and rewards of other users,* neighbors' actions are also a very important context, as they can have not only predictive power but also can impact future rewards through spillover. Moreover, influence susceptibility can vary for different people based on their preferences and the closeness of ties to other users which leads to heterogeneity in the spillover effects. Here, we present a framework that allows contextual multi-armed bandits to account for such heterogeneous spillovers when choosing the best arm for each user. Our experiments on several semi-synthetic and real-world datasets show that our framework leads to significantly higher rewards than existing state-of-the-art solutions that ignore the network information and potential spillover.

## Keywords

recommender systems, multi-armed bandits, information diffusion, social networks

## 1 Introduction

Contextual multi-armed bandit (CMAB) algorithms leverage user attributes and actions to optimize personalized recommendations over time and thus maximize rewards [1, 13, 22, 43]. Rewards can vary based on the application where the recommendations occur, including revenue from recommended products in e-commerce applications and clicks on recommended user-generated content on social media. When contextual bandit recommendations occur in social networks, they can spread from one user to another and overall rewards can be based on both direct recommendations and spillover. Network spillover refers to the phenomenon where the actions of one individual have an impact on the actions of others leading to the spread of information, ideas, attitudes, and behaviors. Understanding the effect of spillover is important in many fields, including psychology [9], marketing [12], public health [33], and economics [35].

To illustrate the motivation behind this paper, let's consider the following toy example: an advertisement company targets social network users with two types of ads, ads on politics and ads on sports. Alice is one of the targeted users and she regularly posts about and engages with fitness-related content. Therefore, when a sport-related ad about downloading a fitness app comes to Alice's news feed, Alice shares the ad link in her social media account. Due to sharing, the ad link appears in the news feed of two of her

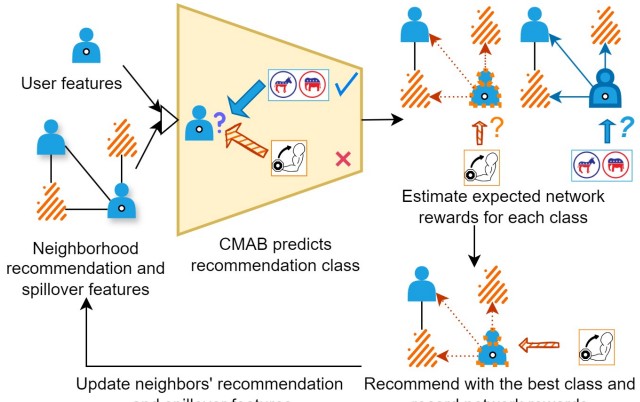

**Figure 1: The workflow of the *NetCB* framework.**

friends, namely Brian and Carla. Brian and Carla are not targeted users but Brian is also interested in fitness and downloads the app from the shared link while Carla is not interested in sports and does not download it. Rewards (i.e., downloads) occur both based on the direct recommendation (Alice's download) and based on spillover (Brian's download), therefore potential spillover should be taken into consideration when recommending items to Alice.

When different users respond differently to the sharing actions of their network contacts, this refers to spillover heterogeneity. For example, it was more likely for Alice to influence Brian to download the fitness app due to their shared interest than to influence Carla. Current research on contextual bandit recommendations [1, 13, 22, 43, 45] does not take into account spillover, much less heterogeneous spillover, to optimize rewards. Thus, in this paper we set out to examine whether heterogeneous spillover can be utilized to maximize the overall rewards during contextual bandit recommendation in social networks.

**Present work.** We develop a **Net**work **C**ontextual **B**andit framework, *NetCB*, that leverages heterogeneous spillover and neighborhood knowledge for recommending items that maximize rewards in networks. As context for the bandit, we introduce a novel dynamic feature set for each user which captures the spillover dynamics over time and provides summary neighborhood statistics about the success of direct recommendations and spillovers. Another novel component of our framework is deciding when to recommend a different arm than the optimal arm predicted by the contextual multi-armed bandit if it leads to higher network rewards due to spillover. Figure 1 illustrates the workflow of the *NetCB* framework. *NetCB* recommends items in rounds where each round corresponds to a user arriving (e.g., opening a social network app). In the first step of each round, the CMAB uses the features to predict the best class to recommend for that user (e.g., politics, marked in blue). Next, *NetCB* checks whether the predicted class is optimal for the neighborhood by comparing the expected rewards due to spillover

for different preference classes. If the network rewards (i.e., direct recommendation and spillover rewards) are highest for a class different from the one the CMAB suggests, then *NetCB* recommends the user that different class instead of the predicted one, going against the CMAB recommendation. In our example, estimated network rewards are higher for the alternate recommendation class, i.e., sports, shown in orange, and the user is recommended that class. In the final step of each round, neighbors' recommendations and spillover features are updated accordingly. *NetCB* can be seamlessly integrated with any existing contextual multi-armed bandit algorithm.

**Key idea and highlights.** To summarize, this paper makes the following contributions:

- We define a novel problem of maximizing *network rewards* for contextual bandit recommendation.
- We introduce a dynamic heterogeneous spillover model and develop a bandit framework which leverages spillover knowledge to increase long-term rewards.
- We perform a thorough evaluation of our framework on semi-synthetic and real-world datasets and compare it to state-of-the-art contextual bandit algorithms.

To the best of our knowledge, this is the first work that considers the impact of both recommendations and their heterogeneous spillover in networks when learning optimal recommendations and calculating bandit regrets. The only research that considers contextual bandits for networks is in the context of influence maximization [19, 36–38] where the goal is to find a set of influential users who will be treated with the goal of these users spreading information in the network. In contrast, our work assumes that anyone could influence others [4] and we focus on the choice of recommendations, not the choice of users. Moreover, unlike previous research on contextual bandit recommendations [10, 22, 26] which leverages only user and item characteristics, we leverage information about the user neighborhood.

## 2 Related Work

**Recommendations with contextual bandits.** Contextual multi-armed bandit algorithms are widely used in recommender systems [10, 26]. LinUCB [22], Contextual Thompson Sampling (CTS) [1], and LinEI [34] algorithms assume a linear relationship between the expected reward of an action and its context. NeuralBandit1 [2], NeuralUCB [45], NeuralTS [43], NeuralEI [34], and EE-Net [7] use neural networks to remove the constraint of linearity. To leverage the collaborative nature among users/items, different algorithms have been developed to model the dependency among items/users, e.g., GOB.Lin [11], CLUB [15], DYNUCB [27], COFIBA [24], CoLin [39], DCCB [20], CAB [14], SCLUB [23], GRC [40], ConUCB [44], DistCLUB [25], LOCB [5], HCB/pHCB [32], Meta-Ban [6], and GNB [28]. However, none of them consider the potential of spillover in maximizing rewards during recommendation.

**Contextual bandit for networks.** The only research that considers contextual bandits for networks is in the context of influence maximization [19, 36–38]. The influence maximization problem aims to maximize rewards in a social network by finding the most influential users that can maximize diffusion in the network. However, influence maximization differs from our work since we focus

on the choice of recommendations to users, not on the choice of users.

**Spillover.** Spillover is typically studied in the context of causal effect estimation in non-iid settings such as social networks and online marketplaces [16, 18, 21, 41]. Some example works in this space include characterization of the neighborhood information through exposure mappings [3] and machine learning [42], as well as using machine learning to estimate heterogeneous effects [8, 42]. All these works focus on spillover to avoid biased estimates of the treatment effect whereas our work focuses on leveraging the heterogeneity of network spillover to maximize rewards in recommendations.

## 3 Problem Description

**Data model.** We define an attributed network graph $G = (V, E)$, consisting of a set of $n$ nodes $V = \{v_1, v_2, \ldots, v_n\}$, a set of edges $E = \{e_{ij}, 1 \leq i, j \leq n\}$ where $e_{ij}$ denotes the edge connecting node $v_i \in V$ and node $v_j \in V$. The set of neighboring nodes of $v_i$ is denoted with $\mathcal{N}_i$ where $\mathcal{N}_i = \{v_j : v_j \in V, e_{ij} \in E\}$. We define $\mathcal{N}_i$ as the neighborhood of node $v_i$ and each node $v_j \in \mathcal{N}_i$ is a neighbor of node $v_i$. Each node in the network has one latent preference from a set of $l$ possible latent preferences (or classes), $C = \{c_1, c_2, \ldots, c_l\}$. In our toy example, the latent preferences are *sports* and *politics*.

We let $\mathbf{X}_i$ denote the $d$-dimensional feature vector of node $v_i$ and $z_i \in C$ refers to its latent preference type. To capture neighborhood properties related to spillover, we introduce a $4l$-dimensional dynamic neighborhood feature set for each node $v_i$ of the network, denoted as $\mathbf{X}_{\mathcal{N}_i}$. This is a novel component of our framework, described in detail in Section 4.1. Let $y_i \in \{0, 1\}$ refer to the activation status of node $v_i$ after a recommendation where $y_i = 1$ means active node (e.g., downloading a fitness app) and $y_i = 0$ means inactive.

The contextual bandit contains a set of arms $\mathcal{A} = \{c_1, c_2, \ldots, c_l\}$ corresponding to user preferences. We denote the predicted by the contextual bandit preference of node $v_i$ with $arm_i \in \{\mathcal{A} \cup \{\emptyset\}\}$ and the recommended class of node $v_i$ (which can sometimes be different from the predicted arm) with $t_i \in \{\mathcal{A} \cup \{\emptyset\}\}$. $arm_i = \emptyset$ refers to no prediction made and $t_i = \emptyset$ refers to no recommendation made to node $v_i$. We provide a detailed description of the bandit setup at the end of this section.

**Node activation.** Nodes in the network can be activated in two ways, through direct recommendation by the system and through spillover. *Direct recommendation* refers to the system treating with a recommendation a particular node ($t_i \neq \emptyset$) in the network. A spillover occurs when the activation of one node impacts the activation of another node. For example, when Alice is shown a fitness app ad that she downloads and shares with Brian, spillover occurs when Brian also downloads the app as a consequence of Alice's sharing.

**Recommendation types.** We define two types of direct recommendations based on the predicted preference class and the actual latent preference of the nodes. The recommendation to node $v_i$ is *aligned* when its recommended class, $t_i$ is the same as its latent preference, $z_i$. Similarly, the recommendation to node $v_i$ is *misaligned* when its recommended class, $t_i$ is different from its latent preference, $z_i$. We denote the probability of activating a particular node due to the aligned and misaligned recommendation as $p_a$ and

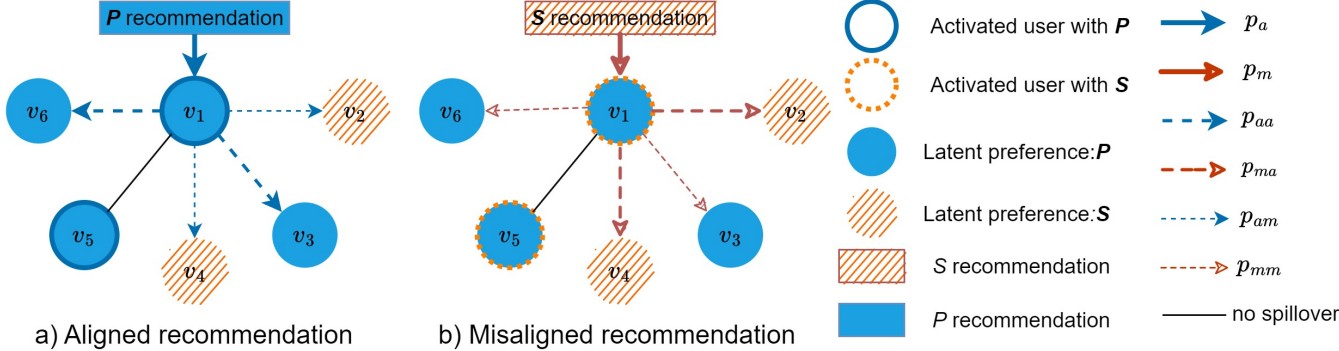

**Figure 2: Recommendation-dependent heterogeneity in network spillover.**

$p_m$, respectively:

$$p_a \leftarrow P(y_i = 1 | t_i = z_i); \qquad p_m \leftarrow P(y_i = 1 | t_i \neq z_i)$$

In our toy example, the activation probability of aligned recommendation would correspond to the probability of Alice downloading the fitness app when she was recommended the fitness app. Similarly, the activation probability of misaligned recommendation would correspond to the probability of Alice acting upon a politics ad when she was shown such an ad.

**Spillover types.** For ease of exposition, we model spillover with the widely used independent cascade model (ICM) [30], though our work can easily be adapted to incorporate other diffusion models. According to ICM, activated nodes have a probabilistic and independent chance of activating an inactive neighbor via spillover. This resembles contagious disease spread, where each social interaction may trigger an infection.

We define an activated node due to direct recommendation as a *source node*. Neighboring nodes that can get activated by the source node are considered *recipient nodes*. In our example, Alice is the source node and Brian is the recipient node. Just like direct recommendations, a spillover can be aligned or misaligned dependent on the latent preference type of nodes and the recommended class of the source node. A spillover is considered *aligned* when the recommended class of an activated source node is the same as that of the latent preference type of the recipient node. Similarly, a spillover is considered *misaligned* when the recommended class of a source node is different from the latent preference type of the recipient node.

We denote spillover probability with $p_{sr}$ where $s \in \{a, m\}$ refers to whether the recommendation of the source node is aligned or misaligned and $r \in \{a, m\}$ refers to whether the spillover is aligned or misaligned. We formulate four types of heterogeneous spillover probabilities where $v_i$ is the source node and $v_j$ is the recipient node:

$$p_{aa} \leftarrow P(y_j = 1 | t_i = z_i, t_i = z_j)$$
$$p_{am} \leftarrow P(y_j = 1 | t_i = z_i, t_i \neq z_j)$$
$$p_{ma} \leftarrow P(y_j = 1 | t_i \neq z_i, t_i = z_j)$$
$$p_{mm} \leftarrow P(y_j = 1 | t_i \neq z_i, t_i \neq z_j)$$

An example of $p_a$, $p_m$, $p_{aa}$, $p_{am}$, $p_{ma}$, and $p_{mm}$ is shown in Figure 2 where a toy network contains six nodes $V \in \{v_1, v_2, v_3, v_4, v_5, v_6\}$ and two possible preferences $C \in \{P, S\}$ (e.g., *politics* and *sports*). Here, $z_1 = P, z_2 = S, z_3 = P, z_4 = S, z_5 = P, z_6 = P$. In Figure 2(a),

$t_1 = P$ and thus $t_1 = z_1$, therefore $v_1$ gets aligned recommendation. $v_3$ and $v_6$ get the aligned spillover from the source node as $t_1 = z_3$ and $t_1 = z_6$, respectively, from the source node $v_1$ that is activated due to aligned recommendation. $v_2$ and $v_4$ get the misaligned spillover from the source node as $t_1 \neq z_2$ and $t_1 \neq z_4$, respectively, from the source node $v_1$ that is activated due to aligned recommendation. In Figure 2(b), $t_1 = S$ and thus $t_1 \neq z_1$, therefore $v_1$ gets misaligned recommendation. $v_2$ and $v_4$ get the aligned spillover from the source node as $t_1 = z_2$ and $t_1 = z_4$, respectively, from the source node $v_1$ that is activated due to misaligned recommendation. $v_3$ and $v_6$ get the misaligned spillover from the source node as $t_1 \neq z_3$ and $t_1 \neq z_6$, respectively, from the source node $v_1$ that is activated due to misaligned recommendation. In both networks, there is no spillover from node $v_1$ to $v_5$ since $v_5$ is already activated and spillover can happen from the currently recommended and activated node to its inactive neighboring nodes.

We assume the heterogeneous spillover probabilities (i.e., $p_{sr}$ where $s \in \{a, m\}$) are known in advance. An interesting follow-up work would be to learn them from data.

**A contextual bandit with network rewards setup.** We consider a stochastic $l$-armed contextual bandit setup with a total number of $T$ rounds. In each round $i \in \{1, 2, 3, \ldots, T\}$, the learning agent receives an inactive node $v_i \in V$ along with a context feature vector: $\{\mathbf{X}_i, \mathbf{X}_{\mathcal{N}_i}\}$. The agent selects an action $arm_i$ and receives network rewards $R(v_i, arm_i)$. The action of an arm is a direct recommendation of an item from one of the classes $C$. A reward refers to the activation of an inactive node due to direct recommendation or spillover. An example reward model is assigning a reward of 1 when a node is activated; otherwise, the reward is 0. The network rewards $R(v_i, arm_i)$ is a non-negative integer which refers to the total number of newly activated nodes due to the selected arm's action, including the node's activation and potential spillover to its neighboring nodes. We assume that the network rewards are a function of the features that needs to be learned: $R(v_i, arm_i) = F(\mathbf{X}_{i, arm_i}, \mathbf{X}_{\mathcal{N}_i, arm_i}) + \xi_i$, where $\xi_i$ is zero-mean noise. The total $T$-round network rewards of the bandit are defined as $R_{total} \leftarrow \sum_{i=1}^{T} R(v_i, arm_i)$. Similarly, we define the optimal $T$-round network rewards as $R^*_{total} \leftarrow \sum_{i=1}^{T} R(v_i, arm^*_i)$, where $arm^*_i$ is the arm with maximal expected network rewards in round $i$. The T-round cumulative regret of the bandit learning can be formulated as $Regret \leftarrow \sum_{i=1}^{T} (R(v_i, arm^*_i) - R(v_i, arm_i))$. We are now ready to formally define the problem:

PROBLEM 1 (MAXIMIZING NETWORK REWARDS WITH CONTEXTUAL BANDITS). *Given an attributed network graph $G = (V, E)$, a set of attributes $\mathbf{X}$ associated with each node, and a set of arms $\mathcal{A}$ associated with the user preferences, select a recommendation class for direct recommendation to each inactive node $v_i$ at round $i \in \{1, 2, 3, \ldots, T\}$ such that the network rewards $R_{total}$ are maximized.*

## 4 Network Contextual Bandit framework

We design a contextual bandit framework that aims to select the optimal direct recommendation for each arriving node in a network by taking into account both the consequences of direct recommendation and indirect recommendations, i.e., spillover. Our **Net**work **C**ontextual **B**andit framework, *NetCB*, comprises two components, considered in each round. The first component (Section 4.1) integrates the use of dynamic neighborhood features in conjunction with the static features of the inactive node to make a prediction for the best direct recommendation. The second component (Section 4.2) leverages heterogeneous spillover and goes against the predicted direct recommendation if it achieves suboptimal expected network rewards. The two components of *NetCB* are described next and the pseudo-code is included in Algorithm 1.

### 4.1 Dynamic neighborhood features per user

Bandit online learning is characterized by the increase in the ratio of successful recommendations to the total number of direct recommendations. To improve the learning, we introduce a set of novel dynamic features which captures the spillover dynamics over time and provides summary neighborhood statistics about the success of direct recommendations and spillovers. This set of features proves to be quite powerful, as we demonstrate in the Experiments section. Specifically, in addition to the static node features $\mathbf{X}_i \in \mathbb{R}^d$, we consider dynamic neighborhood features $\mathbf{X}_{\mathcal{N}_i} \in \mathbb{R}^{4l}$ of node $v_i$.

**1) Neighborhood recommendations per class.** The first $l$ dimensions of $\mathbf{X}_{\mathcal{N}_i}$ represent the ratios of direct recommendations to neighbors $\mathcal{N}_i$ for each of $l$ arms $c_k \in \mathcal{A}$ relative to the total direct recommendations to neighbors for all arms in $\mathcal{N}_i$:

$$\mathbf{X}_{\mathcal{N}_i}[k] = \frac{\sum_{v_j \in \mathcal{N}_i} \mathbb{1}[t_j = c_k]}{\sum_{v_j \in \mathcal{N}_i} \mathbb{1}[t_j \neq \emptyset]}$$

where $k \in \{1, 2, \ldots, l\}$ and $\mathbb{1}[.]$ is an indicator function.

**2) Unsuccessful neighborhood recommendations per class.** The second $l$ dimensions represent the ratios of unsuccessful direct recommendations in $\mathcal{N}_i$, i.e., direct recommendations that fail to activate a node, for each arm $c_k \in \mathcal{A}$ relative to the total unsuccessful direct recommendations for all arms in $\mathcal{N}_i$:

$$\mathbf{X}_{\mathcal{N}_i}[l + k] = \frac{\sum_{v_j \in \mathcal{N}_i} \mathbb{1}[t_j = c_k \wedge y_j = 0]}{\sum_{v_j \in \mathcal{N}_i} \mathbb{1}[t_j \neq \emptyset \wedge y_j = 0]}$$

Let's consider a case from Figure 2(a), where $t_2 = t_5 = t_6 = P$, $t_3 = \emptyset$, and $t_4 = S$. Therefore, $\mathbf{X}_{\mathcal{N}_1}[1] = \frac{3}{4}$ and $\mathbf{X}_{\mathcal{N}_1}[2] = \frac{1}{4}$. If the node $v_5$ gets activated ($y_5 = 1$) due to direct recommendation but all other neighboring nodes of node $v_1$ remain inactive, then $\mathbf{X}_{\mathcal{N}_1}[3] = \frac{2}{3}$ and $\mathbf{X}_{\mathcal{N}_1}[4] = \frac{1}{3}$.

A spillover is *successful* when a recipient node gets activated due to the activation of a source node. A spillover is considered *unsuccessful* when a source node fails to activate the recipient node.

Each potential recipient node $v_i \in V$ has two $l$-dimensional vectors, i.e., $\mathcal{S}_i$ and $\overline{\mathcal{S}_i}$ to keep count of successful and unsuccessful spillovers per preference class, respectively. $\mathcal{S}_i = \vec{0}$ and $\overline{\mathcal{S}_i} = \vec{0}$ indicate that node $v_i$ has received no successful or unsuccessful spillover, respectively. $\mathcal{S}_i[k] \in \{0, 1\}$ refers to whether $v_i$ is activated with $c_k$ through spillover from a neighboring source node. $\overline{\mathcal{S}_i}[k] \in \mathbb{Z}^+$ refers to the total unsuccessful spillover attempts to activate $v_i$ with $c_k$.

**3) Neighborhood spillovers per class.** The third $l$ dimensions of $\mathbf{X}_{\mathcal{N}_i}$ represent the ratios of spillover attempts for each arm $c_k \in \mathcal{A}$ relative to the total spillover attempts in $\mathcal{N}_i$ which can be written as follows for $k \in \{1, 2, \ldots, l\}$:

$$\mathbf{X}_{\mathcal{N}_i}[2l + k] = \frac{\sum_{v_j \in \mathcal{N}_i} (\mathcal{S}_j[k] + \overline{\mathcal{S}_j}[k])}{\sum_{v_j \in \mathcal{N}_i} \sum_{r \in \{1, 2, \ldots, l\}} (\mathcal{S}_j[r] + \overline{\mathcal{S}_j}[r])}$$

**4) Unsuccessful neighborhood spillovers per class.** The fourth $l$ dimensions represent the ratios of unsuccessful spillover attempts for each arm $c_k \in \mathcal{A}$ relative to the total unsuccessful spillover attempts in $\mathcal{N}_i$ which can be written as follows:

$$\mathbf{X}_{\mathcal{N}_i}[3l + k] = \frac{\sum_{v_j \in \mathcal{N}_i} (\overline{\mathcal{S}_j}[k])}{\sum_{v_j \in \mathcal{N}_i} \sum_{r \in \{1, 2, \ldots, l\}} (\overline{\mathcal{S}_j}[r])}$$

Let's consider another case from Figure 2(a), where the two neighboring nodes of node $v_3$ get activated due to "P" direct recommendation and one other neighboring node gets activated due to "S" direct recommendation. Therefore, the node $v_3$ receives spillover with "P" twice and "S" once. Similarly, the nodes $v_4$ and $v_5$ get spillover with "P" once. The nodes $v_1$ and $v_6$ do not receive any spillover. Therefore, $\mathbf{X}_{\mathcal{N}_1}[5] = \frac{4}{5}$ and $\mathbf{X}_{\mathcal{N}_1}[6] = \frac{1}{5}$. If the node $v_5$ gets activated ($y_5 = 1$) due to spillover but all other neighboring nodes of node $v_1$ remain inactive, then $\mathbf{X}_{\mathcal{N}_1}[7] = \frac{3}{4}$ and $\mathbf{X}_{\mathcal{N}_1}[8] = \frac{1}{4}$.

The neighborhood features are dynamic and change over time due to the arrival of new nodes and potential new activations in each round. The aggregated features $X_i$ and $X_{\mathcal{N}_i}$ are passed to an off-the-shelf contextual multi-armed bandit (CMAB) algorithm, which predicts the optimal recommendation class ($arm_i$) for direct recommendation. The CMAB learns the parameters of the arms with the arrival of nodes and generalizes expected network rewards from direct recommendation and spillover.

### 4.2 Spillover maximization

The second part of the algorithm considers the optimal recommendation arm, predicted by the off-the-shelf CMAB, and decides whether to follow the prediction and recommend that arm or go against the prediction and recommend a different arm to the node considered in that round. To do that, it estimates the potential rewards from spillover for all possible arms and picks the arm that gives the highest expected network reward. For example, NetCB may decide to show a politics ad to Alice instead of a sports one (even though she likes sports) if a lot of her inactive friends like politics and the expected network rewards are estimated to be higher. Since we cannot estimate the expected network rewards without knowing the true user preference, we first predict the preferred arm/class for all inactive nodes in the neighborhood and treat them as if they are the true arm (which is unknown). This allows us to

decide which heterogeneous spillover probability applies for each neighbor. The expected network rewards due to the direct recommendation of the predicted recommendation class $arm_i$ for node $v_i$ and spillover in its neighboring nodes is denoted with $E[R_i \mid arm_i]$, which is estimated as follows:

$$E[R_i \mid arm_i] = p_a + p_a * \sum_{v_j \in \mathcal{N}_i} (p_{aa} * \mathbb{1}[arm_i = arm_j \wedge$$
$$y_j = 0] + p_{am} * \mathbb{1}[arm_i \neq arm_j \wedge y_j = 0]) \quad (1)$$

The expected network rewards due to the direct recommendation of each alternate arm, $arm \in \mathcal{A}$ ($arm \neq arm_i$) for node $v_i$ and spillover in its neighboring nodes is denoted with $E[R_i \mid arm]$, which is estimated as follows:

$$E[R_i \mid arm] = p_m + p_m * \sum_{v_j \in \mathcal{N}_i} (p_{ma} * \mathbb{1}[arm = arm_j \wedge$$
$$y_j = 0] + p_{mm} * \mathbb{1}[arm \neq arm_j \wedge y_j = 0]) \quad (2)$$

We denote the alternate arm with highest expected network reward for node $v_i$ with $\overline{arm_i}$, i.e., $\overline{arm_i} = \arg\max_{arm} E[R_i \mid arm]$.

If the expected network rewards of the $\overline{arm_i}$ is greater than that of the $arm_i$, the $NetCB$ framework selects $\overline{arm_i}$; otherwise, it selects $arm_i$ for direct recommendation to node $v_i$. When the alternate arm $\overline{arm_i}$ is selected, the arm parameters are not updated based on the recorded network rewards to avoid ambiguity in off-the-self CMAB bandit learning.

It is important to note that the success of this step depends on having good recommendation predictions for each node. To avoid considering poor predictions in the early learning stages of the CMAB, this step only applies after the direct activation rate (DAR) has stabilized. The $DAR$ refers to the ratio of total activated nodes due to direct recommendations to the total direct recommendations made during contextual bandit learning.

## 4.3 Illustration of NetCB algorithm

Formally, the $NetCB$ algorithm proceeds in discrete rounds $i = 1, 2, 3, \ldots, T$ [Line: 9] following the initialization [Line: $3 - 8$]. In each round $i$, it repeats the two steps described in Sections 4.1 and 4.2.

(1) An inactive node $v_i$ arrives to receive direct recommendation. The algorithm observes the context vector $\mathbf{X}_i$ of the current node $v_i$ along with $\mathbf{X}_{\mathcal{N}_i}$ and bandit parameters for the set of arms, $\mathcal{A}$ [Line: 10].
(2) An off-the-shelf CMAB algorithm predicts an arm, $arm_i \in \mathcal{A}$ for direct recommendation to node $v_i$ [Line: 11].
(3) If $DAR$ is stable, estimate expected network rewards for all arms [Line: 13-15] and find the maximum expected reward generating arm, $\overline{arm_i}$ for node $v_i$ [Line: 16].
(4) If $DAR$ is not stable or the predicted arm $arm_i$ is the same as $\overline{arm_i}$, recommend the predicted class, $t_i = arm_i$ to node $v_i$. Node $v_i$ then receives network rewards, $R(v_i, arm_i)$. The bandit parameters and neighborhood features, $\mathbf{X}_{\mathcal{N}_i}$ are updated. The algorithm then updates its arm-selection strategy with the observation, $(\{\mathbf{X}_i, \mathbf{X}_{\mathcal{N}_i}\}, arm_i, R(v_i, arm_i))$ [Line: 19-20].

---

**Algorithm 1** Maximizing network rewards with $NetCB$

1: **Input:** Number of rounds $T$, off-the-shelf CMAB(e.g., LinUCB, NeuralTS etc.) hyperparameters
2: **Output:** Direct recommendation $t_i$ for each inactive node $v_i$ arrived in each round $i$
3: **for** all $a \in \mathcal{A}$ **do**
4:     Initialize arm parameters
5: **end for**
6: **for** all $v_i \in V$ **do**
7:     $\mathbf{X}_{\mathcal{N}_i}, \mathcal{S}_i, \overline{\mathcal{S}_i} \leftarrow \vec{0}$
8: **end for**
9: **for** $i \in \{1, 2, 3, \ldots, T\}$ **do**
10:     An inactive node $v_i$ arrives with a set of arm contexts $\{\mathbf{X}_{i,a}, \mathbf{X}_{\mathcal{N}_{i,a}}\}_{a \in \mathcal{A}}$
11:     Predict $arm_i$ for node $v_i$ using off-the-shelf CMAB
12:     **if** $DAR$ is stable **then**
13:         **for** all $arm \in \mathcal{A}$ **do**
14:             Estimate expected network rewards of node $v_i$ for $arm$, $E[R_i \mid arm]$
15:         **end for**
16:         Find $\overline{arm_i} = \arg\max_{arm} E[R_i \mid arm]$
17:     **end if**
18:     **if** $DAR$ is not stable OR $arm_i == \overline{arm_i}$ **then**
19:         $t_i \leftarrow arm_i$ # recommend CMAB's prediction
20:         Record $R(v_i, arm_i)$; update parameters for $arm_i$
21:     **else**
22:         $t_i \leftarrow \overline{arm_i}$ # go against CMAB's prediction
23:         Record $R(v_i, \overline{arm_i})$
24:     **end if**
25:     **for** $v_q \in \mathcal{N}_i$ **do**
26:         Update $\mathbf{X}_{\mathcal{N}_q}[k]$, $\mathbf{X}_{\mathcal{N}_q}[l + k]$, $\mathcal{S}_q[k]$, and $\overline{\mathcal{S}_q}[k]$ where $k \in \{1, 2, \ldots, l\}$
27:         **for** $v_r \in \mathcal{N}_q$ **do**
28:             Update $\mathbf{X}_{\mathcal{N}_r}[2l + k]$ and $\mathbf{X}_{\mathcal{N}_r}[3l + k]$ where $k \in \{1, 2, \ldots, l\}$
29:         **end for**
30:     **end for**
31: **end for**

---

(5) If the predicted arm $arm_i$ is different from $\overline{arm_i}$, recommend the direct recommendation class, $t_i = \overline{arm_i}$ to node $v_i$. Node $v_i$ then receives network rewards, $R(v_i, \overline{arm_i})$ [Line: 22-23].
(6) The neighborhood features are updated for each $v_q \in \mathcal{N}_i$ and $v_r \in \mathcal{N}_q$ [Line: 25-30].

The complexity of the algorithm depends on the complexity of the chosen off-the-shelf CMAB and the average degree of the network for updating the dynamic features in each round.

While the NetCB framework is designed to handle the arrival of one node at a time, as per the contextual multi-armed bandit literature, it may also be extended to accommodate batches of nodes arriving simultaneously. In this case, the spillover activation of a recipient node can depend on multiple source nodes trying to activate it.

# 5 Experiments

We evaluate the performance of *NetCB* on both real-world and semi-synthetic network datasets using state-of-the-art CMAB methods showing whether *NetCB* improves these CMAB methods.

## 5.1 Data representation

**Real-world datasets.** We consider four real-world attributed network datasets. BlogCatalog[1] is a network of social interactions among bloggers on the BlogCatalog website. This dataset contains $5,196$ nodes, $343,486$ edges, $8,189$ attributes, and 6 labels. The labels represent topic categories inferred through the metadata of the blogger interests. Flickr[1] is a benchmark social network dataset which contains $7,575$ nodes, $479,476$ edges, $12,047$ attributes, and 9 labels. Each node in this network corresponds to a user, with each edge representing a following relationship, and the labels indicating the interests of groups of the users. The Hateful dataset is sampled from the Hateful Users on Twitter dataset [29] and the sample contains $3,218$ nodes, $9,620$ edges, $1,036$ attributes, and 2 labels. Each sample is classified as either "hateful" or "normal". Pubmed[1] is a citation network where each node represents a scientific publication related to diabetes and each directed edge represents a citation. This dataset contains $19,717$ nodes, $44,338$ edges, $500$ attributes, and 3 labels. Each publication is classified into one of the 3 labels. The *Shannon Equitability Index*[2] values for the Blogcatalog, Flickr, Hateful, and Pubmed datasets are 0.9992, 0.9996, 0.6314, and 0.9651, respectively. As such, all datasets exhibit a high degree of balance, except for the Hateful dataset.

**Semi-synthetic datasets.** We generate semi-synthetic dataset with different homophily for each real-world network dataset. Homophily is quantified by the proportion of edges connecting two nodes with the same label compared to the total number of edges in the network. The homophily scores in Blogcatalog, Flickr, Hateful, and Pubmed network datasets are 0.40, 0.23, 0.73, and 0.80, respectively.

To increase homophily in a network dataset, we employ a random edge removal of edges that connect two nodes with different labels, and where both nodes have a minimum degree of 2. By using this method, we generate semi-synthetic Flickr and Blogcatalog datasets with a homophily of 0.88. To decrease homophily in a network dataset, we employ a random swapping of nodes that have different labels within the network along with their associated attributes and labels. We only consider pairs of nodes where swapping them leads to an increase in the number of edges connecting two nodes with different labels. By using this method, we generate semi-synthetic Pubmed and Hateful datasets with a homophilic score of 0.30 and 0.58, respectively.

**Static features and latent preference of a node.** The static $d$-dimensions in feature vector $\mathbf{X}_i$ correspond to the attributes in the datasets. To reduce computational complexity of the bandit algorithm, we reduce the dimension of $\mathbf{X}_i$ to 500 with truncated SVD [17] for all datasets. The latent preference $z_i$ of node $v_i$ in the network corresponds to its label in the dataset where $l$ refers to the total number of labels. We aggregate $d$-dimensional static node features with $4l$ dimensional dynamic neighborhood features

---

[1]All datasets available at https://renchi.ac.cn/datasets/
[2]The Shannon Equitability Index [31] quantifies class imbalance in a dataset, with 0 being the most imbalanced and 1 being the most balanced.

and consider it as a $d + 4l$ dimensional feature vector for each node in the network. The context feature vector of each arm follow recommendation settings for classification datasets in previous works [7, 22, 28, 43, 45].

## 5.2 Evaluation metrics

**Bandit accuracy.** The bandit accuracy, $B_{acc}$ refers to the ratio of total aligned direct recommendation predictions to the total direct recommendation predictions made during contextual bandit learning, i.e., $B_{acc} = \frac{\sum_{v_i \in V}(\mathbb{1}[arm_i = z_i])}{\sum_{v_i \in V}(\mathbb{1}[arm_i \neq \emptyset])}$

**Regrets.** To evaluate regrets, we record network rewards for node $v_i$ by counting newly activated nodes due to direct recommendation and spillover at round $i$. We compare them to the maximal network rewards for node $v_i$ and report the T-round cumulative regrets defined in Section 3. The maximal network rewards is calculated through simulations.

## 5.3 Main algorithms and baselines

**Baseline CMAB.** We consider several state-of-the-art CMAB algorithms, i.e., LinUCB [22], NeuralUCB [45], NeuralTS [43], and EE-Net [7] both as subroutines in our NetCB framework, and as baselines. We also include GNB [28] which has been shown to have better performance than CLUB [15], DYNUCB [27], COFIBA [24], SCLUB [23], and Meta-Ban [6]. The received reward is 1 if the learning agent predicts the aligned direct recommendation class; otherwise, the reward is 0. Following the literature of CMAB-based recommendation [7, 22, 28, 43, 45] system, we only consider CMAB-based recommendation systems.

**NetCB$_{CMAB}$.** This method accounts only for the first component of our NetCB framework, utilizing dynamic neighborhood features, but not considering going against the bandit recommendation. The received network rewards is passed to the underlying CMAB subroutine of NetCB as implicit feedback which is a non-negative integer. When the underlying CMAB subroutine of NetCB framework is LinUCB, NeuralUCB, NeralTS, EE-Net, and GNB, we denote the methods with $NetCB_{LinUCB}$, $NetCB_{NeuralUCB}$, $NetCB_{NeuralTS}$, $NetCB_{EENet}$, $NetCB_{GNB}$, respectively.

**NetCB$_{\overline{CMAB}}$.** This method accounts for both components of our NetCB framework. The received network rewards is calculated similar to that of NetCB$_{CMAB}$. This version of NetCB does not update parameters in the underlying CMAB subroutine for the received reward when the selected direct recommendation class contradicts the prediction from the first component. When the underlying off-the-shelf CMAB subroutine of NetCB framework is LinUCB, NeuralUCB, NeralTS, EE-Net, and GNB, we denote the methods with $NetCB_{\overline{LinUCB}}$, $NetCB_{\overline{NeuralUCB}}$, $NetCB_{\overline{NeuralTS}}$, $NetCB_{\overline{EENet}}$, $NetCB_{\overline{GNB}}$, respectively.

## 5.4 Experimental setup

We consider single-hop spillover where only immediate neighbors can be activated. We consider a range of possible recommendation and spillover probabilities and show a representative set of results in our experiments using $p_a = 0.7$, $p_m = 0.5$, $p_{aa} = 0.3$, $p_{ma} = 0.3$, $p_{am} = 0.0$, and $p_{mm} = 0.0$ for the first three experiments, and varying them for the others, as specified later. In all of our experiments, we set $p_{mm} = 0$, $p_{am} = 0$. We do conduct a grid search for the exploration constant $\alpha \in \{0.01, 0.1, 0.3, 0.5, 1, 2, 5\}$ of LinUCB. For

**Table 1: Total regrets, at the last round in real-world (white) and semi-synthetic datasets (gray) with standard deviation.**

| Dataset | Flickr | | Blogcatalog | | Hateful | | Pubmed | |
|---|---|---|---|---|---|---|---|---|
| **Homophily** | **0.23** | **0.88** | **0.40** | **0.88** | **0.58** | **0.73** | **0.30** | **0.80** |
| $LinUCB$ [22] | $11097 \pm 686$ | $11536 \pm 641$ | $8421 \pm 375$ | $9160 \pm 797$ | $1574 \pm 89$ | $1496 \pm 49$ | $10862 \pm 104$ | $10993 \pm 315$ |
| $NetCB_{LinUCB}$ | $\textbf{\textit{10012}} \pm 460$ | $8721 \pm 581$ | $\textbf{7040} \pm 267$ | $6026 \pm 350$ | $\textbf{\textit{1520}} \pm 26$ | $1423 \pm 63$ | $\textbf{10817} \pm 159$ | $9186 \pm 208$ |
| $NetCB_{\overline{LinUCB}}$ | $10490 \pm 482$ | $\textbf{8506} \pm 369$ | $7107 \pm 494$ | $\textbf{5856} \pm 301$ | $1602 \pm 38$ | $\textbf{\textit{1409}} \pm 72$ | $10894 \pm 136$ | $\textbf{9163} \pm 133$ |
| $NeuralUCB$ [45] | $11033 \pm 1052$ | $11002 \pm 841$ | $8495 \pm 829$ | $9116 \pm 436$ | $\textbf{\textit{1484}} \pm 78$ | $\textbf{\textit{1538}} \pm 63$ | $\textbf{\textit{11145}} \pm 97$ | $11263 \pm 386$ |
| $NetCB_{NeuralUCB}$ | $12970 \pm 1169$ | $9111 \pm 737$ | $\textbf{7003} \pm 315$ | $5827 \pm 410$ | $1510 \pm 20$ | $1652 \pm 45$ | $11477 \pm 298$ | $9586 \pm 191$ |
| $NetCB_{\overline{NeuralUCB}}$ | $\textbf{\textit{10951}} \pm 806$ | $\textbf{8578} \pm 532$ | $7306 \pm 592$ | $\textbf{5318} \pm 488$ | $1515 \pm 39$ | $1691 \pm 66$ | $11191 \pm 171$ | $\textbf{9371} \pm 390$ |
| $NeuralTS$ [43] | $9590 \pm 272$ | $10301 \pm 461$ | $8177 \pm 263$ | $8604 \pm 753$ | $1619 \pm 85$ | $1647 \pm 43$ | $\textbf{\textit{11322}} \pm 161$ | $11510 \pm 198$ |
| $NetCB_{NeuralTS}$ | $\textbf{\textit{9355}} \pm 307$ | $9326 \pm 222$ | $7220 \pm 265$ | $5974 \pm 325$ | $\textbf{\textit{1604}} \pm 73$ | $1482 \pm 53$ | $11396 \pm 158$ | $10704 \pm 383$ |
| $NetCB_{\overline{NeuralTS}}$ | $9425 \pm 342$ | $\textbf{8626} \pm 544$ | $\textbf{7189} \pm 506$ | $\textbf{5593} \pm 408$ | $1637 \pm 48$ | $\textbf{\textit{1461}} \pm 69$ | $11789 \pm 415$ | $\textbf{10554} \pm 283$ |
| $EENet$ [7] | $\textbf{\textit{9845}} \pm 414$ | $10206 \pm 477$ | $8520 \pm 567$ | $9049 \pm 557$ | $1737 \pm 372$ | $2033 \pm 98$ | $\textbf{\textit{10567}} \pm 183$ | $10375 \pm 315$ |
| $NetCB_{EENet}$ | $12115 \pm 1931$ | $8188 \pm 495$ | $8356 \pm 183$ | $6286 \pm 486$ | $\textbf{\textit{1509}} \pm 47$ | $1937 \pm 122$ | $11026 \pm 200$ | $9350 \pm 146$ |
| $NetCB_{\overline{EENet}}$ | $11523 \pm 707$ | $\textbf{8151} \pm 684$ | $\textbf{6953} \pm 384$ | $\textbf{6128} \pm 293$ | $1532 \pm 37$ | $\textbf{1562} \pm 75$ | $11152 \pm 209$ | $\textbf{9261} \pm 227$ |
| $GNB$ [28] | $9532 \pm 374$ | $10686 \pm 503$ | $8448 \pm 246$ | $9993 \pm 501$ | $\textbf{\textit{1444}} \pm 65$ | $1478 \pm 86$ | $\textbf{\textit{11160}} \pm 142$ | $11217 \pm 298$ |
| $NetCB_{GNB}$ | $\textbf{\textit{9035}} \pm 377$ | $8292 \pm 500$ | $6209 \pm 571$ | $5136 \pm 282$ | $1621 \pm 171$ | $1435 \pm 87$ | $11181 \pm 178$ | $10045 \pm 430$ |
| $NetCB_{\overline{GNB}}$ | $9225 \pm 294$ | $\textbf{8113} \pm 366$ | $\textbf{5786} \pm 234$ | $\textbf{4854} \pm 120$ | $1604 \pm 104$ | $\textbf{\textit{1416}} \pm 94$ | $11423 \pm 166$ | $\textbf{10009} \pm 119$ |

NeuralUCB and NeuralTS, we use a grid search for the exploration parameter $\nu \in \{0.001, 0.01, 0.1, 1\}$, for the regularization parameter $\lambda \in \{0.001, 0.01, 0.1, 1\}$ and for learning rate over $\{0.001, 0.01, 0.1\}$ with a neural network width of 100. For EE-Net [7], we follow their default setup and do the grid search for learning rate over $\{0.0001, 0.001, 0.01, 0.1\}$ for all neural networks. For GNB [28], we follow the default settings for classification dataset in their paper. We choose the best parameters from all grid-searched parameters for each dataset. To determine the stable point, we track the regression slope of direct activation rate (DAR) for the previous $H$ rounds. In each round, the slope is calculated with the $DAR$s of previous $G$ rounds. If the slope remains within a threshold, $|\theta|$, during the previous $H$ rounds, we say the bandit learning, as well as $DAR$, becomes stable, and consequently, we enable our strategy to go against the bandit. Before the start of a bandit experiment, we set $arm_i = \emptyset$, $t_i = \emptyset$, $\mathbf{X}_{\mathcal{N}_i} = \vec{0}$, $\mathbf{S}_i = \vec{0}$, $\overline{\mathbf{S}_i} = \vec{0}$, and $y_i = 0$ for all $v_i \in V$. To determine the stable point for enabling our approach to go against the bandit prediction, we set $G = 300$, $H = 300$, and $\theta = 0.00001$. All experiments are repeated 10 times, and the average results for all methods are reported for comparison. We employ *NVIDIA RTX A5000* GPU on *Ubuntu* 20.04 and Python 3.9.7 to run these experiments.

We run five different experiments. In the first experiment, we look at the effect of dynamic neighborhood knowledge on *Regret* by considering only the first step of NetCB. In the second experiment, we look at the effect of selecting direct recommendation against $NetCB_{CMAB}$ prediction on *Regret* by considering both steps of NetCB. To understand the effect of dynamic neighborhood knowledge on bandit accuracy, $B_{acc}$ learning, we look at the bandit accuracy over time. In the fourth experiment, we look at the effect of activation probability due to direct recommendation on bandit accuracy, $B_{acc}$ with $NetCB_{NeuralTS}$. Finally, we look at the effect of dynamic neighborhood spillover knowledge on the bandit accuracy, $B_{acc}$ in the fifth experiment.

## 5.5 Experimental results

Table 1 shows a summary of the regret comparison between each variant of $NetCB_{CMAB}$ and $NetCB_{\overline{CMAB}}$ with their corresponding CMAB. The best results are shown in bold; ones that are not statistically significantly better are also italicized. The table shows that in most cases (33/40), one of the NetCB variants performs better than its baseline CMAB. In 22 of the 40 cases, one of the NetCB variants has a lower regret that is also statistically significantly better than CMAB. In only 2 of the 40 cases, the baseline CMAB has a better performance that is statistically significant.

*5.5.1 Effect of dynamic neighborhood knowledge on Regret.* NetCB$_{CMAB}$ shows a noticeable decrease in *Regret* compared to its corresponding CMAB baseline, for almost all $NetCB_{CMAB}$ combinations in high-homophily datasets (19 out of 20) and for more than half of the combinations for low-homophily datasets (12 out of 20), as indicated in Table 1. On average $NetCB_{CMAB}$ decreases regret by 17.35% for high-homophily datasets and by 2.47% for low-homophily datasets.

*5.5.2 Effect of selecting direct recommendation against NetCB$_{CMAB}$ prediction on Regret.* NetCB$_{\overline{CMAB}}$ shows a decrease in *Regret* compared to its corresponding CMAB baseline, for 19 out of 20 NetCB$_{\overline{CMAB}}$ combinations in high-homophily datasets and for 10 out of 20 combinations for low-homophily datasets, as indicated in Table 1. On average NetCB$_{\overline{CMAB}}$ decreases regret by 20.15% for high-homophily datasets and by 3.41% for low-homophily datasets. In comparison to NetCB$_{CMAB}$, NetCB$_{\overline{CMAB}}$ decreases regret by 3.52% for high-homophily datasets and by 0.92% for low-homophily datasets.

*5.5.3 Effect of dynamic neighborhood knowledge on the bandit accuracy, $B_{acc}$.* The incorporation of dynamic neighborhood knowledge helps the bandit learn faster in most cases, and thus increases the bandit accuracy, $B_{acc}$ in NetCB$_{CMAB}$s compared to $CMAB$s, particularly in the high homophilic datasets, as shown in Figure 3. NetCB$_{CMAB}$ shows an increase in bandit accuracy, $B_{acc}$ for 19 out

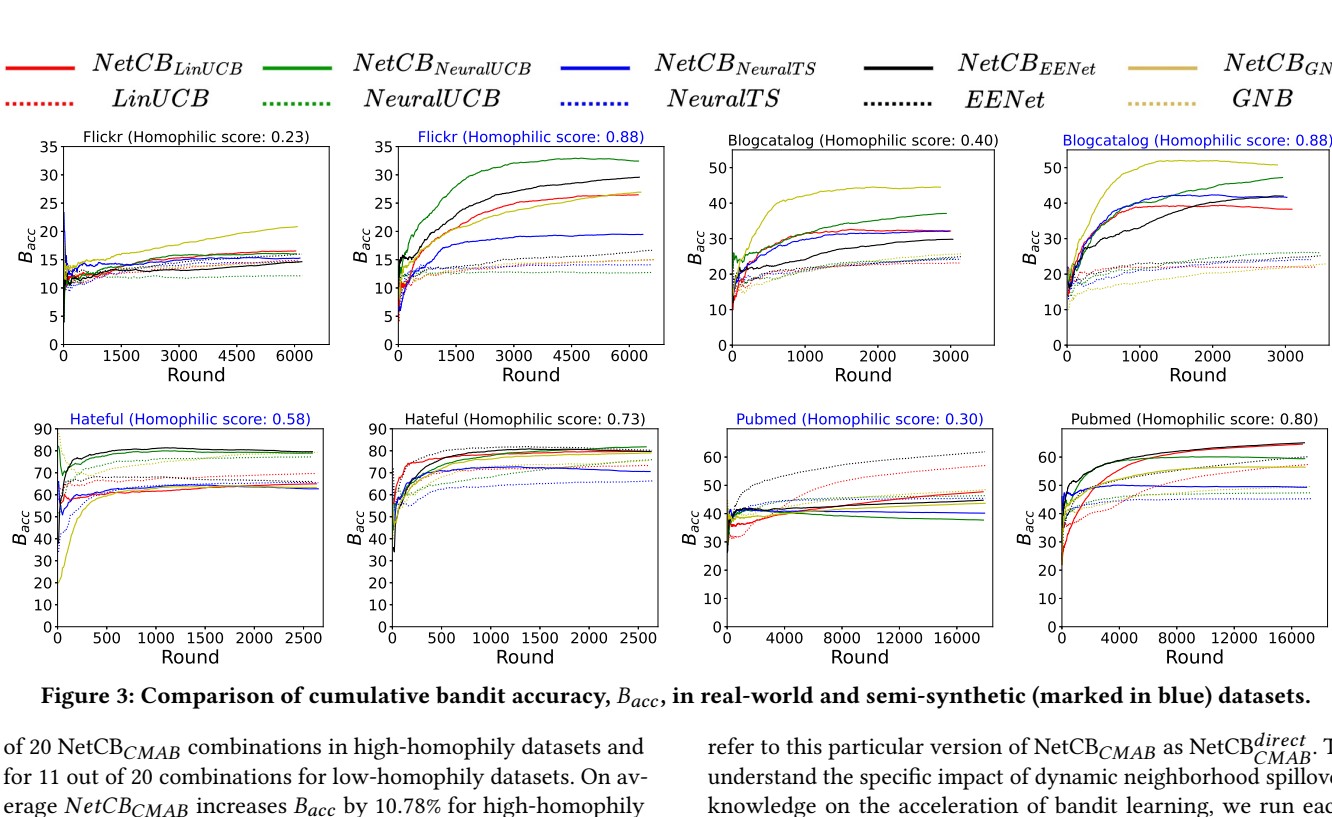

**Figure 3: Comparison of cumulative bandit accuracy, $B_{acc}$, in real-world and semi-synthetic (marked in blue) datasets.**

of 20 NetCB$_{CMAB}$ combinations in high-homophily datasets and for 11 out of 20 combinations for low-homophily datasets. On average $NetCB_{CMAB}$ increases $B_{acc}$ by 10.78% for high-homophily datasets and by 0.56% for low-homophily datasets.

*5.5.4 Effect of activation probability due to direct recommendation on bandit accuracy, $B_{acc}$.* To understand how $B_{acc}$ changes with the increase in difference between $p_m$ and $p_a$, we run $NetCB_{NeuralTS}$ with various combinations of $p_m$ and $p_a$, i.e., (0.5, 0.7), (0.3, 0.7), (0.1, 0.7), in real-world datasets. We run these experiments with a spillover setting of $p_{aa} = 0.3$, $p_{ma} = 0.3$, $p_{am} = 0.0$, $p_{mm} = 0.0$ and observe the cumulative average of the bandit accuracy, $B_{acc}$.

The $B_{acc}$ increases in all datasets as the difference between $p_m$ and $p_a$ increases. Therefore, the bandit can better differentiate among different types of nodes with higher difference between $p_m$ and $p_a$. For example, when $p_m = 0.1$ and $p_a = 0.7$, the bandit achieves around 20.19%, 43.40%, 77.42%, and 56.34% accuracy at the end of experiment in Flickr, Blogcatalog, Hateful, and Pubmed datasets, respectively. However, the bandit achieves 14.73%, 32.10%, 69.82%, and 47.06% accuracy in Flickr, Blogcatalog, Hateful, and Pubmed datasets, respectively, when $p_m = 0.5$ and $p_a = 0.7$. The $B_{acc}$ tends to decrease with the increase in the total number of labels, $l$ of the datasets. For example, the Pubmed ($l = 3$) and Hateful ($l = 2$) datasets achieve higher accuracy than Blogcatalog ($l = 6$) and Flickr ($l = 9$) datasets at the end of the experiments. We show the details of these results in the Appendix.

*5.5.5 Effect of dynamic neighborhood spillover knowledge on the bandit accuracy, $B_{acc}$.* We reduce the dimensions of $\mathbf{X}_{\mathcal{N}_i} \in \mathbb{R}^{4l}$ in NetCB$_{CMAB}$ by removing the last $2l$ dimensions, which represent past knowledge of spillovers in node $v_i$'s neighboring nodes. The resulting $2l$ dimensional representation corresponds to past knowledge of direct recommendations in node $v_i$'s neighboring nodes. We

refer to this particular version of NetCB$_{CMAB}$ as NetCB$_{CMAB}^{direct}$. To understand the specific impact of dynamic neighborhood spillover knowledge on the acceleration of bandit learning, we run each NetCB$_{CMAB}^{direct}$ for real-world and semi-synthetic datasets and compare the results with their corresponding NetCB$_{CMAB}$. We run all these experiments by setting $p_a = 0.7$, $p_m = 0.5$, $p_{aa} = 0.3$, $p_{ma} = 0.3$, $p_{am} = 0.0$, and $p_{mm} = 0.0$. In most experiments with high homophilic networks, dynamic neighborhood spillover information has shown a positive effect on raising the cumulative bandit accuracy, $B_{acc}$, e.g., 4.86% increase in $NetCB_{GNB}$ compared to $NetCB_{GNB}^{direct}$ for Pubmed (Homophily: 0.80) dataset. Nevertheless, the effect is diminished in the lower homophilic networks compared to higher homophilic networks, e.g., 0.12% increase in $NetCB_{GNB}$ compared to $NetCB_{GNB}^{direct}$ for semi-synthetic Pubmed (Homophily: 0.30) dataset.

## 6 Conclusion

We presented NetCB which leverages dynamic neighborhood knowledge and the potential of heterogeneous spillover to maximize network rewards in bandit online learning. Our experiments on real-world and semi-synthetic datasets show a significant decrease in regret when considering neighborhood context in most cases and that it can be beneficial to make suboptimal direct recommendations, in order to maximize rewards from spillover. NetCB can be applied in practical recommendation applications in which the recommendation given to one individual can lead to network rewards when they share that recommendation with their social circles, e.g., video recommendations in social networks, targeted marketing in e-commerce, and healthcare interventions. Future work includes deriving regret bounds for NetCB and automatically learning the values of recommendation-dependent heterogeneous spillover probabilities.

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

# A Appendix

The Appendix contains additional details about the results presented in Section 5.5.

## A.1 Effect of dynamic neighborhood knowledge on Regret

*Regret* in $NetCB_{LinUCB}$, is reduced by 24.40%, 34.21%, 4.87%, and 16.43% when compared to $LinUCB$ in the semi-synthetic Flickr (Homophily: 0.88), semi-synthetic Blogcatalog (Homophily: 0.88), Hateful (Homophily: 0.73), and Pubmed (Homophily: 0.80) datasets, respectively. The datasets show respective decreases of 9.47%, 30.57%, 10.01%, and 7.00% in $NetCB_{NeuralTS}$ compared to $NeuralTS$. In comparison to $EENet$, the datasets show decreases in $NetCB_{EENet}$ of 19.77%, 30.54%, 4.72%, and 9.88%, respectively. The datasets also exhibit respective decreases of 22.41%, 48.60%, 2.9%, and 10.45% in $NetCB_{GNB}$ compared to $GNB$. In these datasets, a reduction in $NetCB_{NeuralUCB}$ is observed in Flickr, Blogcatalog, and Pubmed, with decreases of 17.19%, 36.08%, and 14.89%, respectively, in comparison to $NeuralUCB$.

While the Regret is decreased on average by 17.35% in higher homophilic networks, the impact of incorporating dynamic neighborhood knowledge in reducing *Regret* diminishes in lower homophilic network datasets as shown in Table 1. For instance, all $NetCB_{CMAB}$s generate higher *Regret* compared to $CMAB$s in the semi-synthetic Pubmed (Homophily: 0.30) dataset except in $NetCB_{LinUCB}$, which shows 0.41% decrease in *Regret* compared to $LinUCB$. However, the *Regret* decreases by 9.78%, 2.45%, 5.21% in $NetCB_{LinUCB}$, $NetCB_{NeuralTS}$, and $NetCB_{GNB}$, respectively, when compared to their respective $CMAB$ baselines for the Flickr (Homophily: 0.23) dataset. These decreases in the Blogcatalog (Homophily: 0.40) dataset are 16.4%, 11.71%, and 26.50%, respectively. The dataset also shows a decrease of 17.56% and 1.9% in *Regret* for $NetCB_{NeuralUCB}$ and $NetCB_{EENet}$, respectively, in comparison to their corresponding $CMAB$ baselines. The semi-synthetic Hateful (Homophily: 0.58) dataset indicates a decrease of 3.44%, 0.94%, 13.11% in *Regret* for $NetCB_{LinUCB}$, $NetCB_{NeuralTS}$, and $NetCB_{EENet}$, respectively, compared to their respective $CMAB$ baselines.

## A.2 Effect of selecting direct recommendation against NetCB$_{CMAB}$ prediction on Regret

Our strategy to leverage spillover in the second component helps to decrease the *Regret*s for most $NetCB_{\overline{CMAB}}$s in the datasets with high homophily compared to their respective $NetCB_{CMAB}$s as shown in Table 1. For example, *Regret* in $NetCB_{\overline{LinUCB}}$, is reduced by 2.47%, 2.82%, 0.98%, and 0.25% when compared to $NetCB_{LinUCB}$ in the semi-synthetic Flickr (Homophily: 0.88), semi-synthetic Blogcatalog (Homophily: 0.88), Hateful (Homophily: 0.73), and Pubmed (Homophily: 0.80) datasets, respectively. The datasets show respective decreases of 7.51%, 6.38%, 1.42%, and 1.40% in $NetCB_{\overline{NeuralTS}}$ compared to $NetCB_{NeuralTS}$. In comparison to $NetCB_{EENet}$, the datasets show decreases in $NetCB_{\overline{EENet}}$ of 0.45%, 2.51%, 19.36%, and 0.95%, respectively. The datasets also exhibit respective decreases of 2.16%, 5.49%, 1.32%, and 0.36% in $NetCB_{\overline{GNB}}$ compared to $NetCB_{GNB}$. In these datasets, a reduction in $NetCB_{\overline{NeuralUCB}}$ is observed in Flickr, Blogcatalog, and Pubmed, with decreases of

5.85%, 8.74%, and 2.24%, respectively, in comparison to $NetCB_{NeuralUCB}$.

The strategy to leverage spillover also helps to decrease the *Regret* in some lower homophilic networks, as shown in Table 1. For example, Flickr (Homophily: 0.23) dataset shows a decrease of 15.57% and 4.89% in *Regret* for $NetCB_{\overline{NeuralUCB}}$ and $NetCB_{\overline{EENet}}$, respectively, in comparison to their corresponding $NetCB_{CMAB}$s. The *Regret* decreases by 0.43%, 16.79%, 6.81% in $NetCB_{\overline{NeuralTS}}$, $NetCB_{\overline{EENet}}$, and $NetCB_{\overline{GNB}}$, respectively, when compared to their respective $NetCB_{CMAB}$ in Blogcatalog (Homophily: 0.40) dataset. However, all $NetCB_{\overline{CMAB}}$s generate higher *Regret* compared to $NetCB_{CMAB}$s in the semi-synthetic Pubmed (Homophily: 0.30) dataset except in $NetCB_{\overline{NeuralUCB}}$, which shows 2.49% decrease in *Regret* compared to $NetCB_{NeuralUCB}$. The same goes for the semi-synthetic Hateful (Homophily: 0.58) dataset except in $NetCB_{\overline{GNB}}$, which shows a 1.05% decrease in *Regret* compared to $NetCB_{GNB}$.

## A.3 Effect of dynamic neighborhood knowledge on the bandit accuracy, $B_{acc}$

The bandit accuracy, $B_{acc}$s in $NetCB_{LinUCB}$, increases by 11.48%, 16.31%, 6.45%, and 7.14% when compared to $LinUCB$ in the semi-synthetic Flickr (Homophily: 0.88), semi-synthetic Blogcatalog (Homophily: 0.88), Hateful (Homophily: 0.73), and Pubmed (Homophily: 0.80) datasets, respectively. The $B_{acc}$s of $NetCB_{NeuralUCB}$ increase by 19.70%, 21.15%, 5.84%, and 12.10%, respectively, compared to $NeuralUCB$ in these datasets. The datasets show respective increases of 5.42%, 17.55%, 4.29%, and 4.02% in $NetCB_{NeuralTS}$ compared to $NeuralTS$. In comparison to $GNB$, the datasets show increases in $NetCB_{GNB}$ of 11.88%, 27.87%, 3.49%, and 6.77%, respectively. In case of $NetCB_{EENet}$, the $B_{acc}$ in the Hateful (Homophily: 0.73) dataset decreases by 0.52% compared to $EENet$. However, $B_{acc}$s increase by 12.93%, 16.84%, and 4.93% in the semi-synthetic Flickr (Homophily: 0.88), semi-synthetic Blogcatalog (Homophily: 0.88), and Pubmed (Homophily: 0.80) datasets, respectively, compared to $EENet$.

The impact of dynamic neighborhood knowledge is diminished in lower homophilic datasets compared to higher homophilic datasets as shown in Figure 3. For example, the bandit accuracy, $B_{acc}$ increases by 1.65%, 3.84%, 0.67%, 6.32% in $NetCB_{LinUCB}$, $NetCB_{NeuralUCB}$, $NetCB_{NeuralTS}$, and $NetCB_{GNB}$, respectively, when compared to $LinUCB$, $NeuralUCB$, $NeuralTS$, and $GNB$, respectively, in the Flickr (Homophily: 0.23) dataset. These increases in the Blogcatalog (Homophily: 0.40) dataset are 9.01%, 12.79%, 7.90%, and 18.74%, respectively. All $NetCB_{CMAB}$s yield lower $B_{acc}$s compared to their respective CMABs in the semi-synthetic Hateful (Homophily: 0.58) dataset, except for $NetCB_{NeuralUCB}$ and $NetCB_{EENet}$, which show 1.72% and 13.54% rise relative to $NeuralUCB$ and $EENet$, respectively. The $NetCB_{EENet}$ also shows 4.88% rise relative to $EENet$ in the Blogcatalog (Homophily: 0.40) dataset. All $NetCB_{CMAB}$s generate lower $B_{acc}$s compared to their respective $CMAB$s in the semi-synthetic Pubmed (Homophily: 0.30) datasets.

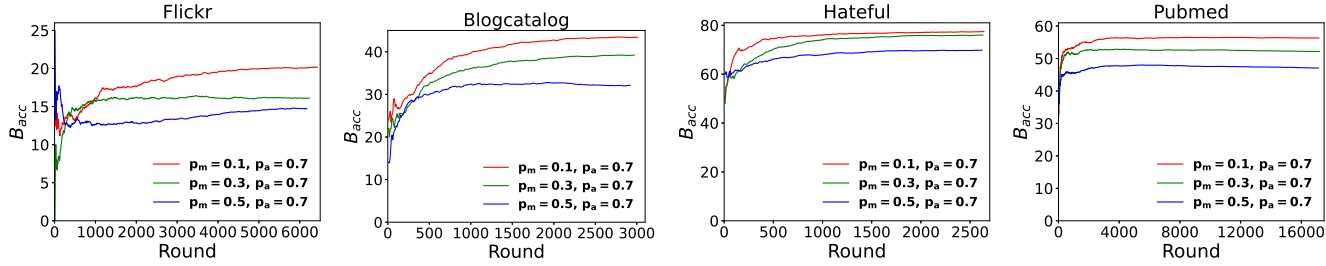

Figure 4: Comparison of cumulative bandit accuracy, $B_{acc}$ of $NetCB_{NeuralTS}$ by varying activation probabilities due to direct recommendations.

## A.4 Effect of activation probability due to direct recommendation on bandit accuracy, $B_{acc}$

The difference in the activation probabilities for aligned and misaligned direct recommendation plays a role in how well the bandit can learn to distinguish between different types of nodes. The bandit learning becomes harder with smaller difference between the activation probabilities as shown in Figure 4. However, neighborhood information helps the bandit learn to distinguish among them,

regardless of the difference in the activation probabilities. In real-world scenarios, $p_m$ and $p_a$ are unknown and therefore it is very important to incorporate neighborhood information.

## A.5 Runtime

The runtime of these experiments depends on the choice of $CMAB$ as well as the density and size of network datasets. For example, $NetCB_{LinUCB}$ requires around 6, 2.5, 4, and 1 hrs to complete a bandit experiment on Flickr, Blogcatalog, Hateful, and Pubmed datasets, respectively. However, the datasets require around four times more hours for $NetCB_{NeuralUCB}$.

