# OpenReview forum: "Leveraging heterogeneous spillover in maximizing contextual bandit rewards"
_ACM.org/TheWebConf/2025/Conference — WWW 2025 Poster_

### Official Review · Reviewer_wHGM · 2024-11-16

**Novelty:** 5
**Technical Quality:** 5

**Review:**

The manuscript introduces a novel framework that integrates heterogeneous spillover effects into the context of contextual multi-armed bandits within social networks.  A series of simulations and experiments on semi-synthetic and real-world datasets have confirmed the findings.  The originality of the work is evident in its novel approach to integrating user preference and spillover effects, which are overlooked in traditional bandit algorithms.  This innovation is significant as it has the potential to enhance the performance of recommender systems by providing more nuanced and personalized recommendations, thereby increasing user engagement. Generally speaking, this manuscript offers some new insights into recommender systems and information diffusion within social networks. However, the manuscript lacks some experimental details and has some formatting issues, such as quantification of heterogeneity, cold-start problems, etc.

**Questions:**

1. The paper assumes that different users respond differently to their network contacts' sharing actions, which is a little ambiguous. Do you mean users with different preferences respond differently? If not, how can we quantify this heterogeneity for users with the same preference?
2. Obviously, this is a simplified model. Current mainstream recommender systems based on deep learning frameworks can finely portray user points of interest and predict user behavior individually. I suggest that the authors test the validity of the model on real data sets and compare it with the baseline model.
3. How does the model handle new nodes joining the network? In the simulation experiment, which spillover probability should the new nodes adopt?
4. In line 691 of page 6, can you explain how these hyperparameters are set? It would be better to provide a hyperparameter sensitivity analysis.
5. The formulas on pages 3 and 4 are not numbered. It would be better to provide a table with the definitions of the notations.

**Reviewer Confidence:**

3: The reviewer is confident but not certain that the evaluation is correct

**Scope:**

4: The work is relevant to the Web and to the track, and is of broad interest to the community

---

### Official Review · Reviewer_brgb · 2024-11-25

**Novelty:** 5
**Technical Quality:** 5

**Review:**

The paper proposes a method for leveraging heterogenous network spillover for maximizing contextual bandits rewards in the case of recommendation systems. In the case of recommendations in a social network, bandit rewards can be maximized by taking into account network spillover rewards, i.e., when someone shares the same recommendation in their social circle and it activates another node. Consequently, (1) dynamic neighborhood features that incorporate bandit recommendation success and neighborhood spillover are fused with the usual user context attributes, and (2) expected network rewards are computed to decide between bandit arm and alternative arms. The presented strategy can be combined with off-the-shelf CMAB algorithms and experimental analysis is shown on several real-world and semi-synthetic datasets.

Quality:
1. Overall, the paper reads well.

Clarity:
1. The exposition is clear and easy to understand.

Originality:
1. The work is not without originality.

Significance:
1. The work is significant in the context of CMAB models by way of leveraging network spillover.

Strengths:
1. The presented framework is simple and easy to understand and can be combined with off-the-shelf CMAB models.
2. The results on various datasets show the efficacy of the presented framework.

Weaknesses:
1. The datasets considered are mostly network classification datasets, but this problem could use other datasets with network spillover effects, such as those used in reco systems and influence maximization works [1,2].
2. The framework doesn't account much for the variation in influence susceptibility of the users- the activation probabilities are hard coded and are the same for every user. Therefore, the heterogeneity in spillover isn't actually addressed.
3. From a recommendations point of view, the approach could be compared with other non-bandit models as well. However, this is also not a major requirement.

[1] A data-based approach to social influence maximization, Goyal et. al., in VLDB12.

[2] Inductive Matrix Completion Based on Graph Neural Networks, Zhang et. al., in ICLR20.

**Questions:**

Please look into the weaknesses stated above. Additionally, I have a few minor questions,
1. How do you decide the sequence of nodes in which to feed the bandit? Does this choice have any effect on the bandit learning? Or is it simply a choice made by the underlying CMAB algorithm?
2. How long does it take to run the simulations to find the maximal rewards for evaluating the total regret?
3. How long does it typically take for bandit learning and the DAR to stabilize?

**Reviewer Confidence:**

3: The reviewer is confident but not certain that the evaluation is correct

**Scope:**

4: The work is relevant to the Web and to the track, and is of broad interest to the community

---

### Official Review · Reviewer_Zisf · 2024-11-27

**Novelty:** 5
**Technical Quality:** 4

**Review:**

This work presents a framework that allows contextual multi-armed bandits to account for such heterogeneous spillovers when choosing the best arm for each user. This is the first work that considers the impact of both recommendations and their heterogeneous spillover in networks when learning optimal recommendations and calculating bandit regrets. The manuscript is generally well-written and innovative, but I have some comments about the work.

**Questions:**

(1) In this manuscript, users affected by spillover will not continue to affect their neighbors, which contradicts the Independent Cascade propagation model. In the IC model, newly activated users will have one chance to activate their inactive out-neighbors. In fact, users affected by spillover may also recommend to their other friends.

(2) Each user seems to have only one preference. Can the model in your paper solve the problem of users having multiple preferences?

(3) In the Experimental Setup Subsection, what is the basis for setting parameters such as $p_a$, $p_m$, $p_{aa}$, $p_{ma}$, etc?

(4) The efficiency of proposed algorithm has not been validated. The authors need to compare the running time of proposed algorithm with other algorithms in the experiments.

**Reviewer Confidence:**

4: The reviewer is certain that the evaluation is correct and very familiar with the relevant literature

**Scope:**

4: The work is relevant to the Web and to the track, and is of broad interest to the community

---

### Official Review · Reviewer_g5gp · 2024-11-27

**Novelty:** 4
**Technical Quality:** 5

**Review:**

This paper proposes a framework called NetCB, which optimises contextual multi-armed slot machine algorithms for selecting the best recommendation items to maximise cumulative rewards in a social network environment by exploiting heterogeneous spillover effects and dynamic neighbourhood knowledge in social networks. To achieve this, this paper designs a dynamic feature set to capture the overflow dynamics. Experimental results show that NetCB significantly improves the performance of the recommender system over multiple datasets compared to existing methods.

**Questions:**

1. How does NetCB determine which nodes are activated because of spillover effects? In other words, how does the dynamic feature set work? I wish the authors could further clarify this section, it's not a friendly read.
2. Can the heterogeneity spillover effect be quantified? How can the authors be sure that the improvement in effectiveness stems from this aspect?
3. What does ‘dynamic neighbourhood knowledge’ mean? It does not seem to appear in the previous section.

**Reviewer Confidence:**

2: The reviewer is willing to defend the evaluation, but it is likely that the reviewer did not understand parts of the paper

**Scope:**

3: The work is somewhat relevant to the Web and to the track, and is of narrow interest to a sub-community

---

### Official Review · Reviewer_Y3mM · 2024-11-29

**Novelty:** 6
**Technical Quality:** 6

**Review:**

The paper presents a well-structured and detailed study on incorporating heterogeneous spillover into contextual multi-armed bandit (CMAB) frameworks. The methodology is thorough, supported by a solid theoretical foundation and rigorous experiments on both synthetic and real-world datasets. The writing is generally clear, and the ideas are logically presented.

The paper explains complex concepts, such as dynamic neighborhood features and heterogeneous spillover, effectively using formulas, diagrams, and examples. However, some dense mathematical descriptions could benefit from further simplification to improve accessibility to a broader audience.

This work addresses a novel problem: maximizing contextual bandit rewards in the presence of heterogeneous spillovers. The introduction of the NetCB framework, which leverages neighborhood-level information, represents a unique contribution compared to existing CMAB approaches. The idea of dynamically modifying recommendations based on network-level spillovers adds significant originality.

The framework has implications for recommender systems, especially in social networks where user influence plays a critical role. By incorporating spillover effects, the approach significantly outperforms traditional CMAB methods in terms of regret minimization and bandit accuracy. This highlights its practical relevance.

Pros:

- Introduces the NetCB algorithm, which integrates dynamic neighborhood features and spillover dynamics.
- Demonstrates effectiveness on various datasets, showing consistent improvements over baselines.
- Addresses real-world scenarios in social networks where user interactions influence outcomes.
- Provides a solid mathematical foundation and clear problem formulation.
- Algorithm design appears scalable, with clear consideration of computational complexity.

Cons:

- Mathematical notations and algorithm descriptions are sometimes overly dense.
- The predefined assumption of spillover probabilities limits the model's adaptability and may reduce its applicability in real-world, dynamic environments.
- Focuses primarily on high-homophily networks, with less analysis of low-homophily scenarios.
- While several CMAB methods are compared, further benchmarks with state-of-the-art reinforcement learning methods could strengthen results.

Overall:

The paper provides a significant contribution to the CMAB literature by addressing the novel problem of leveraging heterogeneous spillover. The proposed NetCB framework demonstrates substantial improvements in both theoretical understanding and practical applications. While the work is impactful and innovative, addressing the noted limitations would further enhance its clarity and practical relevance.

**Questions:**

Can the framework be extended to learn spillover probabilities dynamically from data, rather than assuming they are predefined? If so, how would this affect computational complexity and model performance?

The framework shows stronger performance in high-homophily datasets. What specific challenges does the approach face in low-homophily networks, and how could it be adapted or improved for such scenarios?

Given the dynamic neighborhood feature updates and spillover calculations, how does the computational cost of NetCB scale with larger and denser networks? Are there optimizations to handle real-time applications?

How sensitive is the framework to misaligned recommendations when choosing alternate arms against the CMAB's prediction? Could this decision lead to a degradation in long-term learning?

Have you considered benchmarking NetCB against other modern approaches, such as graph neural networks or reinforcement learning-based methods? How do you position NetCB in comparison to these frameworks?

Could you elaborate on how the model handles cold-start problems for nodes or edges in the network, especially when neighborhood data is sparse or unavailable?

The paper mentions using a threshold for the direct activation rate (DAR) to decide when to go against the bandit’s prediction. How robust is this threshold choice across different datasets, and how sensitive is the framework to its tuning?

The experiments focus on single-hop spillovers. How would the framework handle multi-hop spillover effects, and what would be the trade-offs in terms of performance and complexity?

What are the key challenges in deploying NetCB in a real-world system, and how would you address practical issues such as dynamic user behavior and evolving network structures?

**Reviewer Confidence:**

4: The reviewer is certain that the evaluation is correct and very familiar with the relevant literature

**Scope:**

4: The work is relevant to the Web and to the track, and is of broad interest to the community